# Tiny droplets of ocean island basalts unveil Earth's deep chlorine cycle

Takeshi Hanyu[1], Kenji Shimizu[2], Takayuki Ushikubo[2], Jun-Ichi Kimura [1], Qing Chang[1], Morihisa Hamada[1], Motoo Ito [2], Hikaru Iwamori[1,3,4] & Tsuyoshi Ishikawa[2]

Fully characterising the exchange of volatile elements between the Earth's interior and surface layers has been a longstanding challenge. Volatiles scavenged from seawater by hydrothermally altered oceanic crust have been transferred to the upper mantle during subduction of the oceanic crust, but whether these volatiles are carried deeper into the lower mantle is poorly understood. Here we present evidence of the deep-mantle Cl cycle recorded in melt inclusions in olivine crystals in ocean island basalts sourced from the lower mantle. We show that Cl-rich melt inclusions are associated with radiogenic Pb isotopes, indicating ancient subducted oceanic crust in basalt sources, together with lithophile elements characteristic of melts from a carbonated source. These signatures collectively indicate that seawater-altered and carbonated oceanic crust conveyed surface Cl downward to the lower mantle, forming a Cl-rich reservoir that accounts for 13–26% or an even greater proportion of the total Cl in the mantle.

[1] Department of Solid Earth Geochemistry, Japan Agency for Marine-Earth Science and Technology, Yokosuka 237-0061, Japan. [2] Kochi Institute for Core Sample Research, Japan Agency for Marine-Earth Science and Technology, Nankoku 783-8502, Japan. [3] Earthquake Research Institute, The University of Tokyo, Bunkyo, Tokyo 113-0032, Japan. [4] Department of Earth and Planetary Sciences, Tokyo Institute of Technology, Meguro, Tokyo 152-8550, Japan. Correspondence and requests for materials should be addressed to T.H. (email: hanyut@jamstec.go.jp)

V olatile element distribution in the Earth was influenced by the dynamics and chemical differentiation that operated during the course of the planet's evolution. Drastic out-gassing of the mantle during early Earth history formed a proto-hydrosphere, and this was succeeded by continual volatile outflux from the mantle through volcanism at mid-ocean ridges, ocean islands and subduction zones[1,2]. A return of volatiles from the hydrosphere to the mantle occurs in the scheme of plate tectonics. Subseafloor oceanic crust scavenges volatiles from seawater during hydrothermal reactions prior to subduction[3–5]. Although a greater proportion of volatiles are liberated from subducted oceanic crust beneath fore-arc and sub-arc regions[3–8], recent studies of subduction-zone metamorphic rocks suggest that some volatiles remain and descend to the mantle[3,4]. This theory is supported by the discovery of diamond inclusions containing volatile-rich fluids from subcontinental mantle[9,10] and water-bearing ringwoodite from the mantle transition zone[11]. However, whether volatiles are carried deeper into the lower mantle remains an open question.

Chlorine is suitable for tracing seawater-derived volatiles carried by subducted materials[1,2,4,12–15]. A mantle domain impregnated with seawater-derived volatiles is expected to be enriched in Cl. The volatile composition of the upper mantle has been estimated from studies of mid-ocean ridge basalts (MORB)[15–17]. Enriched MORB mantle sources have higher Cl concentrations (~22 p.p.m.) than normal and depleted MORB mantle sources (5 and 0.4 p.p.m., respectively)[15]. This range of values is thought to reflect heterogeneous Cl distribution in the upper mantle caused by recycling of subducted oceanic crust. In contrast, transfer of Cl-bearing subducted oceanic crust to the lower mantle is poorly constrained because of limited studies on volatiles from ocean island basalts (OIBs) relevant to mantle plumes sourced from the lower mantle[12,13,18,19]. Moreover, magmas erupting in ocean settings are susceptible to Cl contamination by assimilating sea-water, brine and brine-impregnated crust, which obscures the Cl geochemical signature originating from the mantle in both concentration and stable isotope composition[12,15,16,18–20].

A reliable estimate of the Cl content from a mantle source can be obtained through geochemical analysis of olivine-hosted melt inclusions, tiny melt droplets encapsulated in olivine phenocrysts. Volatiles in these melt inclusions are thought to be less modified by secondary assimilation and degassing due to early-stage crystallisation of olivine in the magma. We report Cl and lithophile element compositions combined with Pb isotopes of rehomogenised melt inclusions in OIBs from Raivavae along with existing data from Mangaia[18] in the Austral–Cook Islands, South Pacific. The Raivavae and Mangaia host basalts are characterised by radiogenic Pb isotopic compositions, referred to as high-$\mu$ (HIMU), where $\mu = {}^{238}U/{}^{204}Pb$. This radiogenic Pb signature is believed to have been caused by subtraction of mobile elements, such as Pb, K and Rb, from oceanic crust, while retaining U and Th, during subduction and subsequent radiogenic ingrowth for a few billion years[6,21]. Because seismic tomography demonstrates that the mantle plume relevant to the Austral–Cook Islands chain is sourced from the layer with thermochemical anomaly above the core–mantle boundary[22,23], the subducted oceanic crust stored in the lowermost mantle would be the source of the HIMU basalts at Raivavae and Mangaia[24]. Consequently, whether Cl is delivered to the lower mantle by subducted oceanic crust can be assessed by studying melt inclusions in basalts at these islands.

In this study, we show that Cl enrichment in melt inclusions is associated with radiogenic Pb isotopes, which are indices of subduction-modified ancient oceanic crust in basalt sources. Melt inclusions with radiogenic Pb isotopes also display lithophile element signatures of melts from a carbonated mantle source.

These facts indicate the deep-mantle Cl cycle in which seawater-derived Cl was transferred by subducted altered and carbonated oceanic crust, stored in the lower mantle for billions of years, and returned to the surface through mantle upwelling and partial melting. The Cl enrichment in melt inclusions enables estimation of total Cl amounts subtracted from seawater by subducted oceanic crust, which is equivalent to or even greater than that in the MORB mantle source. We suggest that deep Cl sequestration such as this has played a considerable role in moderating salinity level in the hydrosphere.

## Results

**Pb isotopic compositions.** The Raivavae host basalts have a bimodal Pb isotopic composition[25] (Fig. 1a). The older Rairua Formation basalts have more radiogenic Pb (ie, lower ${}^{207}Pb/{}^{206}Pb$ and ${}^{208}Pb/{}^{206}Pb$), than the younger Anatonu Formation basalts. The Rairua basalts thus contain a larger component from recycled ancient oceanic crust with the HIMU signature than do the Anatonu basalts, which have an isotopic proximity to the MORB mantle source. In situ Pb isotope analyses of melt inclusions indicate that although the Pb isotopic compositions of most of the inclusions overlap the host basalt compositions, some show Pb isotopes intermediate between the Rairua and Anatonu host basalts. Moreover, one Anatonu inclusion notably appears in the Pb isotopic range defined by the Rairua host basalts and vice versa. This suggests localised chemical heterogeneity in the magmas, which is likely caused by mingling of melts from different mantle sources with various Pb isotope compositions. Similar or even greater Pb isotopic variation has been reported in melt inclusions in Mangaia basalts by some authors[26–28], while other authors[18,29] did not find such large variation (Supplementary Fig. 1).

**Elemental compositions.** We determined Cl enrichment by comparing the lithophile elements K and Nb with Cl, which have similar incompatibility during partial melting (Fig. 1b, c)[1,12,15]. After filtering out melt inclusions with anomalously high Cl or unusual lithophile element compositions (5 out of 50; see Supplementary Fig. 2 and Supplementary Note 1), we discovered that the Rairua inclusions have higher Cl/K and Cl/Nb ratios than the Anatonu inclusions and normal MORBs with Cl/K ~ 0.02 and Cl/Nb ~ 5[15], and that these ratios are negatively correlated with ${}^{207}Pb/{}^{206}Pb$ among the Rairua inclusions. Because shallow contaminants, such as seawater, brine, altered oceanic crust and sediments have ${}^{207}Pb/{}^{206}Pb$ similar to or higher than that of the Anatonu inclusions, their assimilation cannot account for Cl enrichment in the Rairua inclusions having radiogenic Pb isotopes (lower ${}^{207}Pb/{}^{206}Pb$). The Mangaia melt inclusions also show high and varied Cl/K and Cl/Nb compared to the Anatonu inclusions and normal MORBs[18]. Chlorine enrichment should thus be a feature of the HIMU mantle source.

Lithophile element compositions also change systematically with Pb isotopes. The Rairua inclusions with the most radiogenic Pb isotopes show prominent depletion in K relative to U, which is ascribed to a preferential loss of K from subducted oceanic crust[6,30] (Fig. 2a). These inclusions also have the lowest $SiO_2$ and highest CaO and $CaO/Al_2O_3$ (Supplementary Fig. 3). They plot outside the field defined by experimental partial melt from carbonate-free peridotite in $SiO_2$–$CaO/Al_2O_3$ space (Fig. 3). Partial melting of carbonate-free silica-deficient eclogite or pyroxenite results in low-$SiO_2$ compositions, as is characteristic of some OIBs, but these partial melts have lower $CaO/Al_2O_3$ than the low-$SiO_2$ OIBs[31]. Therefore, melts from carbonated peridotite and silica-deficient eclogite have been proposed instead as the source of low-$SiO_2$ OIBs[32–34]. This is

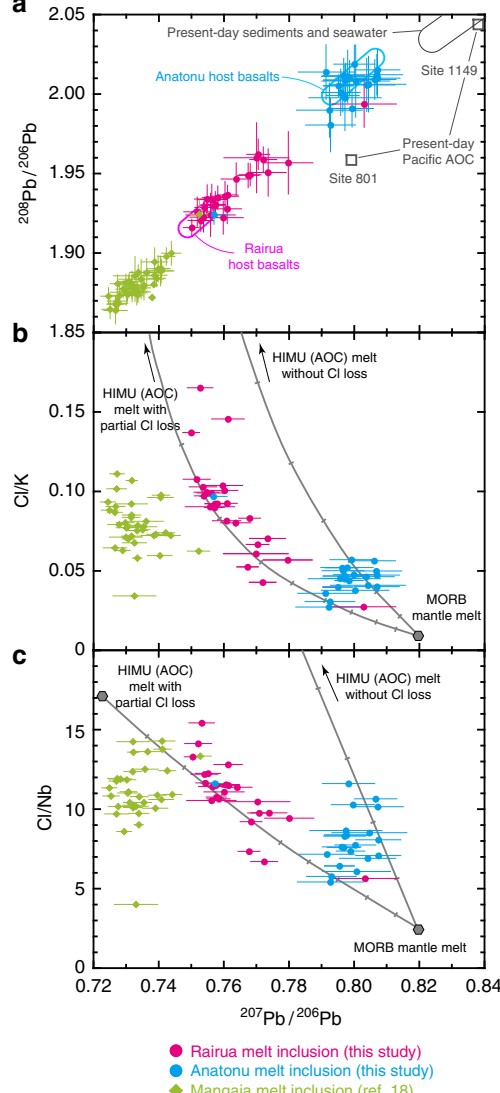

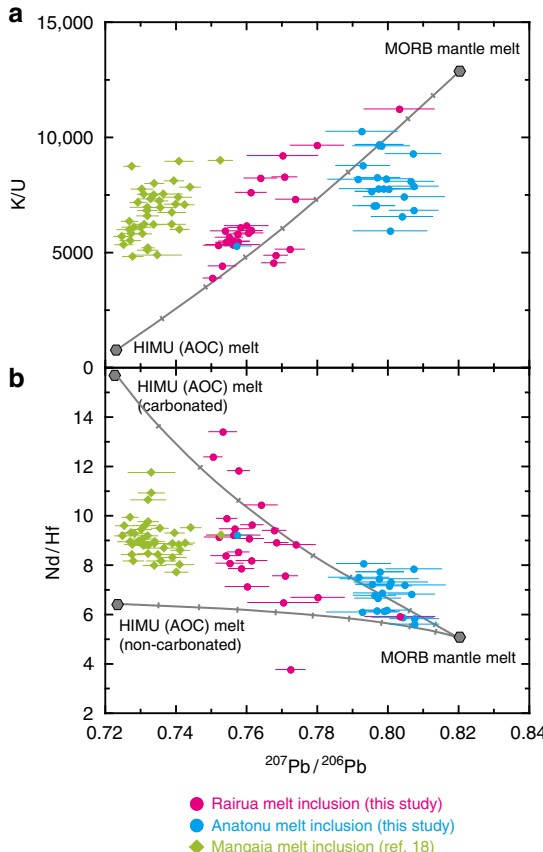

**Fig. 1** Melt inclusion Pb isotopes and Cl-to-lithophile element ratios. **a** $^{207}$Pb/$^{206}$Pb versus $^{208}$Pb/$^{206}$Pb. **b** $^{207}$Pb/$^{206}$Pb versus Cl/K. **c** $^{207}$Pb/$^{206}$Pb versus Cl/Nb. The inclusions in the Rairua and Anatonu basalts from Raivavae are indicated by the purple and light-blue symbols, respectively (this study). For comparison, Mangaia basalt inclusions are indicated by green symbols[18]. Error bars with the $^{207}$Pb/$^{206}$Pb and $^{208}$Pb/$^{206}$Pb data of this study are bracketing standards propagated in-run 2 S.E. (standard errors). The Pb isotopic ranges of Raivavae (Rairua and Anatonu) host basalts are displayed as purple and light-blue fields, respectively, in **a**[25]. Pb isotopic compositions of possible assimilation sources, such as seawater, sediments from the Ocean Drilling Program (ODP) Site 1149, and bulk altered oceanic crust (AOC) from the ODP Sites 801 and Site 1149, are also shown[59]. Mixing lines between melts from the model HIMU source (2 Ga recycled bulk AOC with or without Cl loss during subduction) and MORB mantle source are displayed as grey lines with marks placed at every 10% mixing interval (see Supplementary Note 1 and Supplementary Table 3)

**Fig. 2** Melt inclusion Pb isotopes versus lithophile element ratios. **a** $^{207}$Pb/$^{206}$Pb versus K/U. **b** $^{207}$Pb/$^{206}$Pb versus Nd/Hf. Symbols for melt inclusions are the same as in Fig. 1. Error bars with the $^{207}$Pb/$^{206}$Pb data of this study are bracketing standards propagated in-run 2 S.E. Mixing lines between melts from the model HIMU source (2 Ga carbonated and non-carbonated bulk AOC in **b**) and MORB mantle source are displayed as grey lines with marks placed at every 10% mixing interval (see Supplementary Note 1 and Supplementary Table 3). Melt from the carbonated HIMU source has higher Nd/Hf than that from the non-carbonated one because Hf is less soluble to carbonated melt

consistent with the previous discovery of secondary minerals that formed in equilibrium with carbonate-rich melts in peridotite xenoliths from the neighbouring island of Tubuai[35]. Partial melting of carbonated sources also may have induced the elevated Nd/Hf observed in radiogenic Pb inclusions from the Rairua basalts (Fig. 2b) because of lower partitioning of Hf compared to Nd in carbonated melts[36].

## Discussion

Combined Cl, lithophile elements and Pb isotope data indicate that Cl enrichment in the HIMU mantle source originated from seawater-altered and carbonated oceanic crust that was subducted a few billion years ago. During mantle plume upwelling, this subducted crust was embedded in or reacted with a depleted mantle source, generating a series of melts along binary mixing trends defined by the Rairua and Anatonu inclusions (Figs. 1 and 2). The Rairua and Mangaia melt inclusions have similar Cl and lithophile element compositions, but the Mangaia inclusions have more radiogenic Pb isotopes, possibly reflecting differences in recycling age or extent of (U + Th)/Pb differentiation in the subducted oceanic crust.

We evaluated the efficiency of Cl transfer by subducted oceanic crust by using Cl/Nb ≥ 15 in the HIMU mantle source (Fig. 1c). Here, Cl/Nb was employed as an index of Cl enrichment relative to lithophile elements instead of the more commonly used Cl/K because K is selectively lost from oceanic crust, whereas Nb is one of the conservative elements during subduction. A typical Nb concentration in oceanic crust both before and after sub-arc processes is 2–4 p.p.m.[6–8]. Chlorine is predominantly hosted in altered part of the oceanic crust, which is rich in Cl-bearing minerals, such as amphibole and apatite, and saline fluid

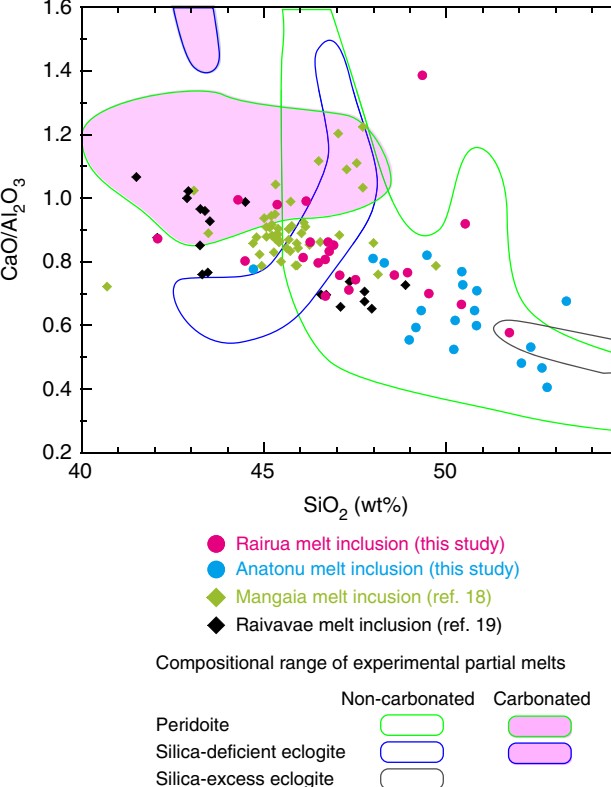

**Fig. 3** $SiO_2$ versus $CaO/Al_2O_3$ of melt inclusions. Symbols for melt inclusions are the same as in Fig. 1. Raivavae inclusions without Pb isotope data from previous studies are indicated by black symbols[19]. The compositional range of experimental partial melts of peridotite, silica-deficient eclogite and silica-excess eclogite are indicated by the fields outlined by green, blue and grey lines, respectively. Experimental partial melts from carbonated sources are shown by purple fields[24, 31–33]

inclusions in nominally anhydrous minerals[2,37–40]. The average Cl concentration in the 7 km section of oceanic crust is estimated to be 78–334 p.p.m.[2,37,38]. Assuming the initial composition of bulk oceanic crust to be 207 p.p.m. Cl and 3.5 p.p.m. Nb (Cl/Nb ~60; Supplementary Table 3)[7,38], the predicted composition of mixed melts from the recycled oceanic crust and MORB source mantle would have Cl/Nb that is too high for the Rairua and Mangaia inclusions (Fig. 1c). Subducted altered oceanic crust could have lost Cl as a consequence of breakdown of Cl-bearing minerals and fluid inclusions during prograde metamorphism under high pressure–temperature conditions in the subducted slab[3,39–41]. This should result in decreasing Cl/Nb in the dehydrated oceanic crust. Despite significant uncertainty, Cl/Nb in dehydrated oceanic crust can be estimated as 7–40, using Cl concentrations reported in ophiolitic eclogites that experienced subduction modification at Syros (28–60 p.p.m.)[42] and Monviso (71–79 p.p.m.)[41]. This overlaps with the Cl/Nb of subducted oceanic crust that formed the HIMU mantle source, implying that the nominal Cl remaining after subduction modification or resupplied by fluid flux from underlying serpentinised lithospheric mantle[1,4,14] could have been conveyed along with the residual oceanic crust to the lower mantle (Fig. 4).

Chlorine-bearing phases so far have not been identified in subducted oceanic crust beyond sub-arc depth. Major minerals in eclogites may be capable of accommodating Cl at lattice defects, although this mechanism seems less efficient because of large ionic radius and anionic character of Cl[43–45]. Alternatively, mineral grain boundaries have been suggested as significant storage sites for elements that are incompatible with major minerals[46]. These incompatible elements including Cl in subducted oceanic crust could be delivered directly to the deep mantle if the slab geotherm is cold. However, recently it was suggested that warm slab geotherms may encounter the solidus of carbonated eclogites in the mantle transition zone (Fig. 4)[47]. According to this alternative explanation, the incompatible element signatures are transferred by carbonatitic melts from the oceanic crust to the overlying metasomatised mantle that is eventually foundered to form the deep HIMU mantle source[48].

Subducted oceanic crust rich in Cl would influence the Earth's Cl budget. Using Nb concentrations (2–4 p.p.m.) and Cl/Nb ($\geq 15$), we estimated the minimum range for Cl concentration of the subducted oceanic crust to be 30–60 p.p.m. It is greater than that of the depleted and normal MORB mantle sources (0.4 and 5 p.p.m., respectively)[15], which together comprise more than half of the silicate mantle[1,49]. Given a constant oceanic crust subduction rate of $5.2 \times 10^{13}$ kg yr$^{-1}$ for 4 Ga[30], the amount of subducted Cl should be at least $0.6$–$1.2 \times 10^{19}$ kg, which is equivalent to or even greater than that in the MORB mantle source ($0.11$–$1.4 \times 10^{19}$ kg; Fig. 4; Supplementary Table 4). The previous estimates of Cl abundance in the whole mantle show a large uncertainty ranging from $0.15 \times 10^{19}$ kg to $14 \times 10^{19}$ kg[50]. If the recent estimate of bulk Earth Cl concentration of 26 p.p.m. is accepted[1], Cl abundance in the mantle after subtracting Cl presently in surface reservoirs ($5.8 \times 10^{19}$ kg; Supplementary Table 4) should be $4.6 \times 10^{19}$ kg. Consequently, subducted oceanic crust may account for at least 13–26% of the total Cl in the mantle. Although a portion of subducted oceanic crust would be dispersed into the upper mantle and mantle transition zone, giving rise to the enriched MORB mantle and subcontinental mantle sources, the remaining portion could have been isolated in the deep mantle, thus forming the Cl-rich HIMU mantle source.

The deep Cl cycle, which is driven by slab subduction, may moderate salinity conditions in the Earth's surface layers. Assuming a constant Cl flux rate of $2$–$5 \times 10^9$ kg year$^{-1}$ (ref. [1]), the volcanic output from MORB and OIB to surface reservoirs is estimated to be $0.8$–$2 \times 10^{19}$ kg Cl over 4 Ga. However, a salinity change model suggests that seawater in the Archaean and Proterozoic was 1.5–2 times more saline than it is today based on the estimate of on-land evaporites ($1.4 \times 10^{19}$ kg Cl) and brine deposits ($1.5 \times 10^{19}$ kg Cl)[50] sequestered from the ocean as the continents grew[51]. The amount of Cl ever subducted to the deep mantle is comparable to that in the evaporites and brines, which implies that the subducted oceanic crust has been a major Cl sink that compensated for mantle outflux and buffered Cl in the hydrosphere to achieve a low-salinity surface environment suitable for life[51].

## Methods

**Sample preparation**. The rock samples from Raivavae are alkali basalts bearing large (1–3 mm) phenocrysts of olivine ± clinopyroxene. We separated olivine phenocrysts from seven Rairua basalts and eight Anatonu basalts from Raivavae. Fresh olivine grains containing melt inclusions were handpicked under a binocular microscope. Forsterite numbers for the host olivine range from 77 to 86. In many cases, the melt inclusions have spheroidal shapes and the typical sizes of the studied inclusions are 50–150 μm on the longest axis (Supplementary Table 1; Supplementary Figs. 4 and 5). The melt inclusions, which are at least partly devitrified, consist of glass, silicates (clinopyroxene, hornblende and olivine crystallised on inclusion rims), oxides (Fe–Ti oxide and spinel), sulphides and vapour bubbles[19]. We rehomogenised the inclusions by reheating and quenching them using a heating stage (Linkam; TS1500). Prior to conducting our heating experiments, individual olivine grains were mounted on acryl resin and doubly polished without exposing inclusions on the polished surface. After removing the resin using acetone, the olivine grains were heated under an Ar atmosphere and monitored under a microscope until the inclusions were completely molten. After maintaining homogenisation temperatures (typically 1180–1280 °C) for 5 min, the samples were quickly quenched by pulling the sample plate from the heating furnace. After reheating, rehomogenised melt inclusions were glassy and light brown in colour

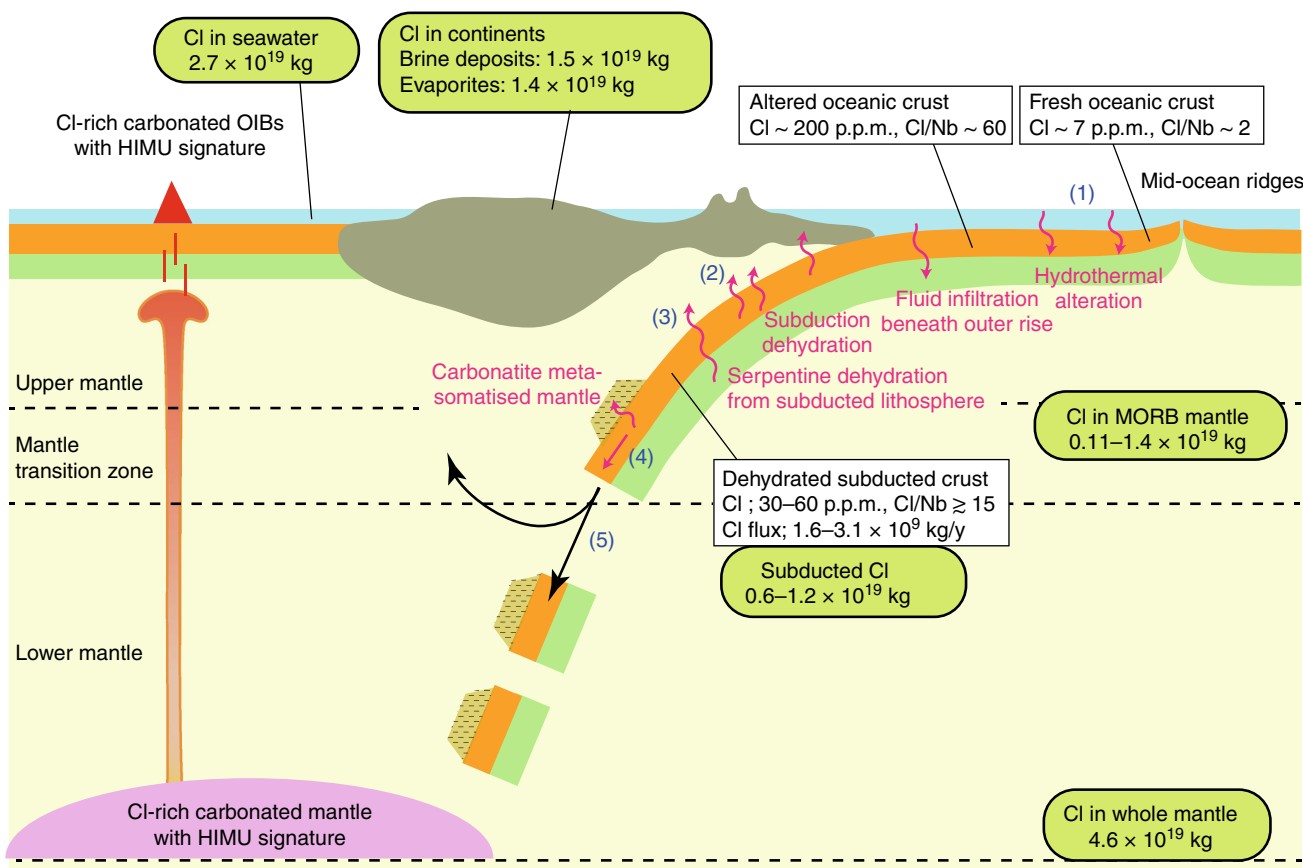

**Fig. 4** Schematic illustration of Cl behaviour during subduction and Cl inventory. White boxes show Cl concentration and flux in transporting media from the surface to the mantle. Green ovals show the Cl inventory in major Cl reservoirs (Supplementary Table 4). Chlorine is transferred by subducted oceanic crust in a way that (1) oceanic crust scavenges Cl from seawater during hydrothermal alteration and that (2) much Cl is lost from subducted crust by breakdown of hydrous minerals, but nominal Cl would remain in major minerals or at mineral grain boundaries after subduction dehydration. (3) An alternative mechanism for Cl transport would be if dehydrated oceanic crust is fluxed by Cl-rich fluids from underlying serpentinised lithospheric mantle that has been hydrated by fluid infiltration through crustal fractures at the outer rise. (4) Incompatible elements, including Cl, would be transported directly by subducted oceanic crust or be imprinted on the metasomatised mantle by oceanic crust-derived carbonatitic melts. (5) A portion of subducted oceanic crust or metasomatised mantle might be dispersed into the shallow mantle, but the remaining portion could have been foundered to form the Cl-rich HIMU mantle reservoir. The minimum estimated Cl transported by dehydrated oceanic crust ($0.6–1.2 \times 10^{19}$ kg) is comparable to the Cl amount in major Cl reservoirs, such as MORB mantle source, crustal brines and evaporites

(Supplementary Figs. 4 and 5). We discarded melt inclusions that burst or became crosscut by cracks during heating. The olivine grains were mounted on acryl resin again and polished to expose rehomogenised glassy melt inclusions on the surface. After removing the resin using acetone, the grains were mounted on indium discs and soaked in acetone and milli-Q water to clean the sample surface.

**Analytical methods**. We determined the compositions of Cl, major and trace elements and Pb isotopes in melt inclusions by a combination of in situ analytical methods using secondary ion mass spectrometry (SIMS), electron probe micro analysis (EPMA) and laser ablation-inductively coupled plasma-MS (LA-ICP-MS). The measured elemental and isotopic compositions are shown in Supplementary Table 1.

**SIMS**. The indium discs were coated with gold and the Cl concentrations were determined using SIMS (CAMECA; IMS 1280HR) at the Kochi Institute for Core Sample Research, the Japan Agency for Marine-Earth Science and Technology (JAMSTEC)[52,53]. A 20 kV (10 kV at the ion source and −10 kV at the sample surface) primary $Cs^+$ ion beam of 300–700 pA was defocused to a 15 μm diameter spot on the sample surface. Presputtering was conducted for 20 s to remove potential surface contaminants before making measurements. Secondary ions from an area of $5 \times 5$ μm at the centre of the spot were transferred to the ion optics using a field aperture. After separating the ions using a mass resolving power of ~6000, secondary ions were measured with an axial electron multiplier using a magnetic peak switching. The $^{35}Cl$ beam intensity was measured and normalised with $^{30}Si$ to determine Cl concentrations. Calibration lines for Cl were determined prior to the sample runs using international and in-house glass standards (Supplementary Fig. 6). In our previous assessment on the reproducibility, the relative standard

deviation of Cl measurements was <3% determined by repeated measurements of various standard glasses (97–2517 p.p.m. Cl) as unknown samples[53]. The analytical uncertainty for Cl is approximated by this value, as it is much higher than in-run precision (Supplementary Table 1). The accuracy of the reported data, which depends on the uncertainty of Cl concentrations of the glass standards used for calibration lines, was estimated to be <6% (1 $\sigma$) during the determination of Cl concentrations of the glass standards by bulk analysis using pyrohydrolysis and ion chromatography (Supplementary Table 2)[53,54].

**EPMA**. After SIMS analyses, the indium discs were polished to remove their gold coatings. After rinsing with acetone and Milli-Q water, the indium discs were dried in a vacuum chamber and coated with carbon for major element analyses using EPMA (JEOL; JXA-8500F) at the Yokosuka Headquarters, JAMSTEC. Major element compositions were determined with a 15 kV accelerating voltage and a 12 nA beam current. The host olivines enclosing the melt inclusions were also analysed using a 20 kV accelerating voltage and a 25 nA beam current.

**LA-ICP-MS**. Trace element compositions of melt inclusions were determined using LA-ICP-MS at the Yokosuka Headquarters, JAMSTEC[55]. A 200 nm femtosecond laser (OK Laboratory; OK-Fs2000K) coupled with a sector-field ICP-MS (Thermo Fisher Scientific; Element XR) was used for the measurements. The ablation pit was set to a diameter of 20–30 μm and a depth of 20 μm. Surface contamination is typically negligible after careful surface cleaning, as described above. Visible signals from surface contamination, if any, were discarded using time-resolving analysis. For normalisation, a GSD-1G basalt standard glass from the United States Geological Survey (USGS) was measured after every five unknown samples. For quality control, a BHVO-2G international glass standard from the USGS was repeatedly

measured (Supplementary Table 2). Standard glass reference values are from the GeoReM database[56].

Finally, Pb isotope ratios were determined using the same 200 nm femtosecond laser coupled with a multiple collector-ICP-MS (Thermo Fisher Scientific; Neptune)[57,58]. The ablation pit was set to a diameter of 30 μm and a depth of 20 μm. To eliminate Pb surface contamination, pre-ablation was performed for 10 s. We avoided cracked parts, former shrinkage bubbles, spinel inclusions and edges of melt inclusions for all measurements except for the samples RAV-08 OL-10 MI-01 and RAV-20 OL-07 MI-01 for which laser spots for Pb isotope analysis partially overlap the surrounding host olivines because otherwise there was not enough room for LA. The signal intensity and isotope ratios were monitored during each sample run by time-resolving analysis. When unusual change in signal intensity or isotope ratios was encountered, such data were discarded. The $^{206}Pb$, $^{207}Pb$ and $^{208}Pb$ beam intensities were simultaneously measured using multiple Faraday collectors equipped with a high-gain $1 \times 10^{13}$ Ω resistor. Other isotopes, such as $^{200}Hg$, $^{202}Hg$ and $^{204}Pb$, were monitored using Faraday collectors with a normal $1 \times 10^{11}$ Ω resistor. An SRM 612 synthetic glass standard from the National Institute of Standards and Technology (NIST) was measured before and after each unknown sample for external mass bias correction by applying a bracketing method. The measurements were performed on two separate days, and, for quality control of the data, the USGS-supplied BCR-2G and BHVO-2G international basalt glass standards were measured five times each day prior to analysing unknown samples (Supplementary Table 2). All reference values for the glass standards are from the GeoReM database[56]. We found that the range of Pb isotope ratios observed in unknown samples was far greater than the measured isotopic variation of the SRM 612 bracketing standard during the sample runs (Supplementary Fig. 7).

**Host rock description.** Basalts from the Austral–Cook Islands chain show diverse isotopic compositions[21,25,60]. Among these islands, Mangaia, Rurutu, Tubuai and Raivavae have been recognised as locations where robust HIMU geochemical signatures (ie, radiogenic Pb isotopes) occur. Raivavae basalts can be divided into two groups, Rairua Formation and Anatonu Formation basalts, based on field observations, ages and chemical compositions[25,61,62]. Rairua Formation basalts have K-Ar ages between 7.4 and 10.6 Ma, and are characterised by radiogenic Pb isotopes with $^{206}Pb/^{204}Pb \geq 20.5$. Anatonu Formation basalts have ages between 5.4 and 6.4 Ma, and are characterised by Pb isotopes that are less radiogenic ($^{206}Pb/^{204}Pb \leq 20.1$). Isotopic compositions of the host basalts that we used in this study are presented previously[62]. Large isotopic variation is a common feature shared by other volcanic islands in this hotspot chain. For example, subaerial and submarine basalts on the neighbouring island of Tubuai show bimodal Pb isotopic compositions[63].

**Correction for post-entrapment crystallisation.** Olivine-hosted melt inclusions from Raivavae are internally nonhomogeneous due to the presence of daughter minerals that crystallised from the original melts when the host basalts experienced slow cooling. Melt inclusions can be further differentiated by olivine overgrowth on the wall of the melt inclusions[64]. In principle, such post-entrapment crystallisation can be reversed by reheating and subsequent quenching of inclusions. However, in some cases, when we applied reheating, it was not sufficient to dissolve all the olivine overgrowth, which are difficult to recognise by an optical observation under a microscope. Conversely, inclusions could have been overheated, causing excess olivine from the surrounding olivine crystals to dissolve[65]. We computed original melt inclusion compositions by determining the proportion of olivine added to or subtracted from the inclusions, ie, the correction factor. Our calculation scheme is conventional: starting with olivine that is in Fe/Mg equilibrium with the melt, we iteratively added it to or subtracted it from inclusions until the calculated Fe/Mg of the inclusions reached equilibrium with the surrounding host olivine[66]. The major element composition of the host olivine was determined using an EPMA. We assumed the Fe/Mg partition coefficient between olivine and melt to be 0.3[67]. The correction factor is within 20% for all studied melt inclusions and within 5% for most of them. The chemical compositions of the inclusions (Supplementary Table 1) were reported after correcting for these effects, assuming that olivine is free of elements other than Mg, Fe and Si. The change of $SiO_2$ content due to the correction is within 0.5 wt% for most of the melt inclusions.

## Data availability
All data generated during this study are included in this published Article and its Supplementary Data files.

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

## Acknowledgements

We thank Kyoto Fission-Track Co. Ltd. for their preparation of olivine separates. Careful review by M.G. Jackson has greatly improved the manuscript. This work was supported by JSPS KAKENHI grant numbers JP26400527, 17H02994 (T.H.), JP15H02148, and JP16H01123, JP18H04372 (J.-I.K.), and 15H03751 (K.S.)

## Author contributions

T.H., H.I. and T.I. designed the study. T.H., K.S., T.U. and M.I. performed SIMS analyses. M.H. contributed to EPMA analyses. T.H., J.-I.K. and Q.C. conducted LA-ICP-MS analyses. T.H., K.S., T.U. and J.-I.K. wrote the draft of the manuscript with contributions from all authors.

## Additional information

**Competing interests:** The authors declare no competing interests.

