## [Peer Review File · Nature Communications]

Reviewer #2 (Remarks to the Author):

Review of paper entitled "Tiny droplets of ocean island basalts unveil Earth's deep chlorine cycle," by Takeshi Hanyu et al.

The authors present a new and important dataset on olivine-hosted melt inclusions from the volcanic island of Raivavae (Austral Islands). They report major, trace, and volatile element concentrations and Pb-isotopic ratios in the individual melt inclusions. After excluding the inclusions with high Cl/Nb, they find a relationship between Cl/Nb and Pb isotopes, and they argue that the HIMU endmember in the inclusions has moderately elevated Cl/Nb (Cl/Nb = 15). They use the observation of moderately high Cl/Nb in the most HIMU inclusions to argue for a Cl-rich HIMU component in the mantle, which they argue is altered oceanic crust (with high Cl/Nb) that has been cycled through the mantle and melted beneath the Austral island of Raivavae. They suggest that subducted altered oceanic crust is Cl-rich, and that the altered oceanic crust does not lose its full complement of Cl during subduction zone processing.

I like this paper. In fact, I have always wanted to do this exact project on John Lassiter's Raivavae melt inclusions, to see if the Pb isotopic compositions correlate with the high Cl/Nb component. So I'm glad that Hanyu et al have done this. A very impressive analytical effort on the part of Hanyu et al.

My primary concern, as detailed below, deals with how the authors filter the inclusions based on their Cl/Nb ratios. It appears that they filter the inclusions with high Cl/Nb because they have high Cl/Nb (there is no other apparent geochemical reason for filtering these inclusions). After filtering the high Cl/Nb inclusions, they argue that the inclusions with the highest Cl/Nb that survived the "Cl/Nb filtering process" must represent the Cl/Nb ratio of the HIMU domain. However, the Cl/Nb ratio of the surviving inclusions will depend on the threshold Cl/Nb value that the authors selected for the filtering process. This seems a bit circular. Can't the authors use another geochemical criteria for filtering? Don't the highest Cl/Nb inclusions have some other geochemical signature that can be used for filtering?

Overall, a really nice manuscript. If the authors deal with the comments below, this manuscript will be suitable for publication. I suggest minor revision is warranted. I do not need to re-review this manuscript.

Matt Jackson

August 2018

UCSB

Specific comments:

1. Line 83. The Raivavae host basalts show a bimodal Pb isotopic composition⁸ (Fig. 1a). Older Rairua Formation basalts have more radiogenic Pb, thus having lower $^{207}\text{Pb}/^{206}\text{Pb}$, than younger Anatonu Formation basalts.

Comment: Explain what you means by having “more radiogenic Pb”. Add something like “i.e., higher $^{206}\text{Pb}/^{204}\text{Pb}$ ”. Both ^{207}Pb and ^{206}Pb are “radiogenic”, so this is potentially confusing.

2. Line 86: The Rairua basalts thereby contain more melts from recycled ancient oceanic crust with the HIMU signature than the Anatonu basalts that have an isotopic proximity with the MORB mantle source.

Comment: “The Rairua basalts thereby contain a larger component from recycled oceanic crust...”

3. Line 92. This suggests that small-scale chemical heterogeneity is present in the magmas, which is caused by mingling of melts from different mantle sources with varying Pb isotope compositions, as previously documented in melt inclusions in Mangaia basalts (Extended Data Fig. 1)

Comment: The two early studies (Yurimoto et al; Saal et al) found extreme heterogeneity for Pb isotopes in Mangaia melt inclusions. But two later studies (Paul et al; Cabral et al) did not. This should be addressed in the text, as this is a concern for the community.

4. Add melt inclusion sizes to Supp Figure 1. Also, please add photos of all melt inclusions. The authors no doubt have such photos, and I suspect this information will be useful to the community as folks evaluate which types of inclusions, based on morphology, are most representative.

5. Line 96. We determine Cl enrichment by comparing Cl with lithophile elements with similar incompatibility during partial melting (i.e., K and Nb)^{9,14,17} as shown in Figs. 1b and 1c. After filtering out melt inclusions with anomalously high Cl (4 out of 52; Extended Data Fig. 2 and Supplementary Information), we discovered that the Rairua inclusions show higher Cl/K and Cl/Nb ratios than the Anatonu inclusions and normal MORBs, with Cl/K ~ 0.02 and Cl/Nb ~ 517 , and that for the Rairua inclusions these ratios are negatively correlated with $^{207}\text{Pb}/^{206}\text{Pb}$.

Comment: This seems circular. See comment below referring to the supplement. After eliminating the high Cl/Nb inclusions, for no clear reason other than the fact that they have high Cl/Nb, the authors argue that one basalt series has higher Cl/Nb than the other. But isn't this a product of the Cl/Nb “threshold value” of 15 that the authors chose? What if the authors chose a lower value, say, Cl/Nb = 5, as their threshold value? This needs to be better justified.

6. Line 103. Shallow contaminants have less-radiogenic Pb isotopes, thus assimilation of seawater, brine, or brine-impregnated oceanic crust is not responsible for Cl enrichment coupled with radiogenic Pb.

Comment: It is important to show this. Please show all possible contaminants in a plot of $^{207}\text{Pb}/^{206}\text{Pb}$ vs $^{208}\text{Pb}/^{206}\text{Pb}$. Show brines, seawater, altered oceanic crust, sediment, etc. Hauff et al. (G-cubed 2003; doi:10.1029/2002GC000421) have some excellent examples of highly altered oceanic crust that would be useful in this regard. All of the possible contaminants should be shown together with the melt inclusions from this study, so that the reader can identify whether the melt inclusions have interacted with these assimilants. This will help the reader understand the arrows in Figure 1, which point to an "assimilation trend". There are so many possible assimilants that the reader is wondering why the assimilation points up and to the right in the plots?

7. Line 104. Despite small Pb isotopic variation, the Mangaia melt inclusions also show high Cl/K and Cl/Nb 19,

Comment: Actually, Yurimoto et al, and Saal et al., found enormous Pb isotopic variability at Mangaia. This is a concern.

8. Line 104. Despite small Pb isotopic variation, the Mangaia melt inclusions also show high Cl/K and Cl/Nb 19, which overlaps that of the Rairua inclusions. Chlorine enrichment should thus be a feature of the HIMU mantle source.

Comment: In Fig 1, a couple of the Mangaia inclusions show low Cl/K and low Cl/Nb, but have the same Pb isotopic compositions as the other Mangaia inclusions. It would be good to mention this. Could it be possible that the low Cl/Nb inclusions from Mangaia are the most "primitive" (i.e., least contaminated)?

9. Line 108. The isotopic variation defined by Raivavae host basalts is best explained by binary mixing of melts from HIMU and MORB mantle sources

Comment: There is another depleted components in the Cook-Australis. Konter et al. (epsl, vol 275, 2008) showed a pervasive "C" component throughout the Cook-Australis. Why do the authors prefer a MORB component over the already-identified C component? This is worth mentioning.

10. Line 112. We calculated the composition of the MORB mantle melt assuming 1% fractional melting of the MORB mantle with bulk partition coefficients between melt and garnet peridotite

Comment: Why do the authors use such a low melt fraction? Please explain (in the text) why such a low value is selected.

11. Cl/Nb ratios in the most HIMU glasses (that is, the 3 most HIMU lavas from Dredge 10) examined by Jackson et al (G-cubed 2015) are 12.8–13.7. These values are inconsistent with the hypothesis put forward by the authors here.

12. Figure 3. It would be useful to add fields for carbonated peridotite and carbonated eclogite/pyroxenite experimental compositions. As it stands, the authors show data trending outside of the “peridotite melting” field, but this doesn’t really illuminate the authors’ preferred model of carbonate-rich melting.

13. Supplement. We did not find any highly Cl-enriched melt inclusions equivalent to type-3 inclusions in our sample sets. However, a small number of inclusions in the Rairua and Anatonu basalts show Cl enrichment that deviate from the compositional trend defined by the majority of inclusions, which was presumably caused by brine assimilation. A single Rairua inclusion had much higher Cl/Nb ratio (~53) than other inclusions (Extended Data Fig. 2a). Three Anatonu inclusions had high Cl/Nb (>15) that define a steep trend in Cl versus Cl/Nb space. This excessively high Cl/Nb is not due to Nb depletion because these inclusions show higher Cl/Nb than the others for a given 1/Nb (Extended Data Fig. 2b). We excluded these four melt inclusions and also some inclusions in the literature (shown by open symbols in Extended Data Fig. 2) from the discussion as they were likely contaminated by secondary brines.

Comment: The authors exclude the high Cl/Nb inclusions because they have high Cl/Nb? Is there another geochemical indicator in these high Cl/Nb inclusions that can be used?

14. Line 108. The isotopic variation defined by Raivavae host basalts is best explained by binary mixing of melts from γ HIMU and MORB mantle sources^{8,43,44}.

Comment: This seems circular, somehow. By excluding the lavas with the highest Cl/Nb (i.e., >15), the authors then conclude in the manuscript that HIMU lavas have Cl/Nb of about 15. Is there some other geochemical indicator that the authors can use to exclude the highest Cl/Nb inclusions? Do the high Cl/Nb inclusions have some other bizarre or strange trace element (Rb/Nb? K/Nb?) or major element (K/Ti?) compositions? It just seems strange to filter using an arbitrary cutoff for Cl/Nb to then conclude that there is a specific Cl/Nb that is “primitive”, as this “primitive ratio” will always be determined based on the arbitrary cutoff used by the authors.

15. Supplementary Figure 3: Try plotting Ca/Al instead of Ca in the lower panel. This usually tends to show better correlations with Pb isotopes, and is also interpreted to reflect a role for carbonatite in the melt source (see Jackson and Dasgupta).

16. Supplementary Table 2. Paul et al and Cabral et al found very little Pb isotopic variability in Mangaia melt inclusions, which contrasts with extreme Pb isotopic variability in the melt inclusions from Mangaia identified by Saal et al and Yurimoto et al. This is a big concern in the community, particularly because Cabral et al worked on olivine-hosted inclusions from two of the same basalts examined by Saal. Therefore, it is critical to document standard and unknown analyses to ensure that any Pb isotopic variability is real, and not an analytical artifact. The following should be addressed explicitly in the supplement to help readers assess the data:

A. It appears that the Pb isotopic standard was run only 5 times during the analytical session?

B. Over how many days were the melt inclusions analyzed? How many analyses of the Pb isotopic standard were made during each day that melt inclusions were analyzed?

C. Documenting the analytical session. A plot of $^{207}\text{Pb}/^{206}\text{Pb}$ and $^{208}\text{Pb}/^{206}\text{Pb}$ vs time, showing melt inclusions and standards, would be extremely useful. Include in run error bars for each analysis, and document different analytical sessions in the plot.

D. Did the laser spot overlap with phase boundaries? Olivine-melt inclusions, or spinel-melt inclusions. Cabral et al found that, whenever the beam overlapped with phase boundaries, they would find Pb-isotopic variability that overlapped with the extreme inclusions identified by Saal et al. It is critical to avoid analyses where the beam overlaps with phase boundaries.

E. Paul et al found that surface contamination on the samples was rich in Pb, and they had to carefully identify and eliminate the data from the first, Pb-rich, pulse of data at the start of each laser analysis. This surface contamination issue is critical to avoid, and should be discussed in the supplement.

Reply to the comments from Reviewer #1

We are grateful for your careful reading of our manuscript and thoughtful comments and suggestions. Following the suggestions by you and another reviewer, we have revised the manuscript. Below, we reply to your comments and explain how we have revised the manuscript.

Reviewer #1; Summary:

These results could be important to the community, as volatile cycling in the mantle is poorly understood. However, the authors fail to present compelling evidence for: 1. Where Cl is stored in their deep subduction model and why that parcel must come from the lower mantle and 2. The nature of AOC as it relates to HIMU melts. Both could be addressed in more detail to improve the manuscript. To the latter, the first author of this submission was a co-author on a recent paper, which used trace element analyses in olivine phenocrysts to conclude that the HIMU source is in fact carbonitite-metasomatized sub-continental lithospheric mantle. How does the author reconcile this HIMU model with the one described in this submission (namely subducted oceanic crust)?

The exchange of volatiles between Earth's surface and silicate mantle has been debated as one of primary issues in mantle geochemistry. We believe that this manuscript could contribute to the study field by presenting an evidence of deep Cl cycle recorded in melt inclusions from HIMU OIBs. As you pointed out, however, the carrier of volatiles in the subducted oceanic crust is poorly understood. Regarding Cl, high-pressure experiments demonstrate that some hydrous minerals in sediments, oceanic crust, and serpentinised subducted lithosphere could deliver Cl to the sub-arc depth (up to ~300 km), but mineral phases that host Cl at deeper levels have not been reported. On this occasion, we discussed in the revised manuscript possible ways for Cl to be transported further deep, following your suggestions. We note that nominal Cl may be accommodated at lattice defects, but also that mineral grain boundaries are significant sites for such element with large ionic radius and anionic character by referring to some previous literatures (Dalou et al., 2012; Joachim et al., 2015; Hauri et al., 2006; Hiraga et al., 2004) (Lines 170-175 in the revised manuscript; please also see the reply below).

The place where the geochemical heterogeneity is preserved in the mantle is another issue of interest. Because the depth of the plume source cannot be always constrained

by geochemistry, we introduced outcome given by mantle seismic tomography in the revised manuscript. Recent seismic studies (Suetsugu et al., 2009; French and Romanowicz, 2015) demonstrate that the mantle plume beneath the Austral-Cook island chain is a primary plume that originated from the core-mantle boundary. Moreover, seismic anomaly at the core-mantle boundary region cannot be explained solely by temperature, and thereby it must include materials that are chemically distinct from the surrounding mantle. We described it in the Introduction (Lines 74-80).

We recognise that there are two major models regarding the way of recycling of altered oceanic crust (AOC). One model is that AOC survives in its form in the deep mantle, is transported by an upwelling mantle plume, and melts/reacts with ambient peridotitic mantle to form OIBs. Another model is that AOC melts/reacts to form fertile peridotite or pyroxenite before reaching to the deep mantle reservoir (Weiss et al., 2016, in which the first author of the present manuscript is a co-author). Both models can explain Cl enrichment in the HIMU OIBs discovered in the present study, because incompatible elements including Cl are mobilized by silicate and/or carbonated melts from subducted oceanic crust, and accordingly the incompatible element signatures could be imprinted on the metasomatised mantle. We discussed this issue (Lines 175-181) with a schematic illustration in Fig. 4 in the revised manuscript.

The paper is organized and concise, and the data appears to be of high quality. I do have some concerns about uncertainty propagation; please see my note regarding SIMS analyses and duplicate samples.

In response to your concerns on the uncertainties for Cl measurements with SIMS, we estimated the uncertainties and reproducibility of the Cl calibration curves in the 'Methods' in the revised manuscript. We also described the accuracy of the measurements there (Lines 251-258). Regarding the duplicate measurements, it was our mistake to show the duplicates in the data table, which was corrected in the revised manuscript. Please see the details in the reply to your comments below.

The manuscript could be improved by addressing the current schools of thought on the origin of HIMU and what, physically, it could represent- and how this present work is relevant to that debate. The paper could use a summary figure in the fourth slot, possibly an illustration of the various reservoirs with Cl sources, sinks, and estimates of the Cl abundance of each. This would tie their salient points together in a clear way for

the reader. To further improve the manuscript, I suggest that the authors expand on the major element composition of the melt inclusions to test if these are consistent with the HIMU components defined in the literature.

In response to your comment, we presented a new Fig. 4 in the revised manuscript that gives a schematic explanation about the transport of Cl from the surface via subducted oceanic crust, Cl concentration before and after dehydration, and the major reservoir for Cl in the mantle and surface layers. We believe that this figure helps readers understand the significance of the subducted oceanic crust as one of major Cl reservoirs in the Earth. We reinforced the argument on the major elements. This was done by using $\text{CaO}/\text{Al}_2\text{O}_3$, rather than CaO, which is more widely used for the discussion of the source lithology (Lines 124-130). We compared compositions of the studied melt inclusions with those of experimental melts not only from silicate and carbonated peridotite (as in the previous manuscript) but also silicate and carbonated eclogite in the revised Fig. 3. We believe that the discussion on the source lithology and the presence of carbonated sources became more comprehensive in the revised manuscript.

Detailed Comments

-Line 53. Both cited papers refer to the mass transfer of volatiles from the surface to the mantle, however they do not provide a mechanism to reach the transition zone, as stated. Page et al. (2016) discusses phengite and amphibole stability to ~300 km in cold subduction zones, while Kendrick et al. (2017) does not mention subduction of serpentinite, or any associated phase change, into the transition zone. I recommend removing “transition zone” from the sentence.

As you pointed out, previous authors showed the possible delivery of volatiles to ~300 km, but did not discuss the mechanism of volatile transport further deep after decomposition of major hydrated phases. Following your suggestion, we removed ‘transition zone’ from the sentence and simply described that some volatiles can be transported beyond the sub-arc depth to the ‘mantle’ (Line 44).

-Line 62. “Enriched MORB mantle has higher Cl concentration (~22 p.p.m) than normal and depleted MORB mantle (0.4–5 p.p.m.)....” Better to list as “normal and depleted MORB mantle (5 and 0.4 p.p.m., respectively)”.

We rephrased the sentence following your suggestion (Lines 52-54).

- Line 110. The authors suggest that CaO-SiO₂ systematics require a carbonated source. I would argue that the data could also represent a garnet pyroxenite melt, e.g. Hirschmann et al. (2003)⁴. Would recommend additional citations.

Thank you for the constructive comment. In the revised manuscript, we changed the plots from the SiO₂-CaO space to the SiO₂-CaO/Al₂O₃ space in the revised Fig. 3, because the latter is more widely used in the previous literatures. In this figure, we showed the compositional ranges of experimental silicate and carbonated melts from not only peridotite but also eclogite (silica-rich and silica-deficient) by referring to Hirschmann et al. (2003), Gerbode and Dasgupta (2010), Dasgupta et al. (2010) and Jackson and Dasgupta (2008). In addition to the previous argument that carbonate-free peridotite cannot generate low-SiO₂ (<45 wt%) alkalic melts, we discussed other possible sources. Carbonate-free silica-deficient eclogite/pyroxenite has been extensively studied and is suggested to be a potential source of low-SiO₂ melts, although the experimental melts have lower CaO/Al₂O₃ (because of too aluminous composition) at low SiO₂. We described accordingly that carbonated sources are more likely to account for the major element compositions of the low-SiO₂ melt after the discussion by Dasgupta et al. (2010) and Jackson and Dasgupta (2008) (Lines 124-130).

-Line 132. I would refer the authors to Barnes et al. (2018)⁵ for additional data on AOC halogen abundances; these could better constrain their Cl/Nb ratios (currently 20 to 170), e.g. Barnes and Cisneros (2012), 207 µg/g Cl. This citation could prove useful in the manuscript.

I agree that Barnes and Cisneros (2012) provided a valuable data set of Cl by studying drill cores from oceanic crust, and that the literature data are compiled in Barnes et al. (2018). We cited the two papers in the revised manuscript. We used the average Cl concentration in the bulk oceanic crust (207 p.p.m.) from Barnes and Cisneros (2012) as a model Cl composition of bulk oceanic crust in the discussion (Line 156 and Supplementary Table 3).

-Line 144. A handful of studies⁶ on natural eclogites suggest that saline fluid inclusions (>10% wt Cl) could host significant quantities Cl during subduction of anhydrous eclogite; how that Cl is redistributed during diffusional re-equilibration (if at all) is still unknown. I suggest adding a citation to that effect.

In response to the comment, we described that saline fluid inclusions are one of major hosts of Cl in the altered oceanic crust in the shallow depth, but that they might be remobilised during prograde metamorphism by referring to Svensen et al. (2001) in the revised manuscript (Lines 152-154; 159-162).

-Line 144. The study of Debret et al. (2016) defies experimental studies⁷, which have shown that Cl is present in sub- $\mu\text{g/g}$ concentrations in Cpx and garnet. This finding is also consistent with lattice strain theory of Cl partitioning, based on Cl's large ionic radius. I would therefore not include this citation. I would suggest that the authors also consider the possibility that Cl could be hosted in grain boundaries, e.g. Hiraga et al.⁸

Thank you for the constructive comment. Debret et al. (2016) reported relatively high Cl concentration in clinopyroxene and garnet in eclogite. However, experimental studies show that although nominal Cl may be accommodated at lattice defects, Cl is less partitioned to minerals because of large ionic radius and anionic character of Cl. We described it by referring to Hauri et al. (2006), Dalou et al. (2012), and Joachim et al. (2015) in the revised manuscript. Alternative sites for Cl in the circumstances where Cl-bearing minerals are absent are, as you pointed out, mineral grain boundaries. We also discussed this possibility by referring to Hiraga et al. (2004) in the revised manuscript (Lines 170-175).

-**Table 3** outlines the melting model parameters used, however there is considerable detail missing. It appears that the authors used the mineral partition coefficient data from Stracke et al. (2003, G3) as applied by Hanyu et al. (2013). Please include mineral partition coefficients, the melting modes used, and a citation to the original Stracke et al. paper so that readers may more easily reproduce the results.

We revised Supplementary Table 3 to show the data sources for model composition and partition coefficients. In response to your comments below (regarding mixing lines in Figs. 1 and 2), we show the compositions of mixing components (i.e., MORB mantle and subducted oceanic crust) after Salters and Stracke (2004) and Becker et al. (2000). Partition coefficients during melting of MORB mantle and oceanic crust are given assuming garnet peridotite from Stracke et al. (2003) and eclogite from Stracke and Bourdon (2009), respectively. The partition coefficients and the mode of minerals in garnet peridotite and eclogite to calculate bulk partition coefficients are also presented

in Supplementary Table 3.

-Table 3. The authors use 1% non-modal fractional melts of depleted MORB mantle to simulate the Cl/K and Cl/Nb content of MORB melts in their model presented in Fig. 1. If these melts were indeed partially sourced from an upwelling mantle plume, such a low extent of garnet peridotite melting is unlikely (authors use 1% melting of a garnet peridotite to define DMM melt end-member). The authors should consider higher extents of melting, then test if those extents are compatible with not only elemental ratios (which will remain similar if compatibilities are similar), but also absolute abundances of various elements in the melt inclusions to provide more convincing evidence.

We assessed if the melting/mixing model reproduces the elemental ratios (Cl/K, Cl/Nb, K/U, and Nd/Hf) of the melt inclusions by changing the melting degree of the depleted source (MORB mantle). Because the elements studied here are incompatible, the mixing trajectories are not largely modified if we assume small (1%) and large (5%) degree of partial melting. We did not assess if such melting/mixing model reproduces the absolute abundance of various trace elements (e.g., spidergram) in the melt inclusions because these melt inclusions are slightly differentiated ($\text{MgO} < 8 \text{ wt}\%$). In order to model the trace elements, it is desired to use the chemical composition of melts with high MgO ($> 8 \text{ wt}\%$), and actually we did it in our previous paper (Hanyu et al., 2013) using the bulk rock composition of high-MgO basalts with radiogenic and less-radiogenic Pb isotopes from Tubuai, the neighboring island of Raivavae. We suggested that low-degree partial melting of the depleted component is requisite to reproduce the composition of heavy rare earth elements. We believe that such low-degree melting was achieved if the depleted component (ambient mantle) was heated near the outer edge of the mantle plume. Moreover, it is suggested (e.g., Stracke, Chemical Geology, 2012) that the depleted peridotitic component would melt at small degree because of the high solidus temperature relative to the recycled eclogite/pyroxenite/fertile peridotite. We recognise that there are uncertainties in chemical compositions, mineral modes and melting degrees of the source components, but for the purpose of model calculation, we assumed 1% non-modal fractional melting of the depleted component (MORB mantle). We described this point in the supplements (Lines 143-154 in the Supplementary Information).

Fig. 2. To define the isotopic and chemical composition of the HIMU source melts, the

authors take the highest $^{207}\text{Pb}/^{206}\text{Pb}$ values and Cl/Nb in their melt-inclusion suite. This is a circular argument, as mixing will generally fall along the trend line. To augment their argument, I would suggest that the authors calculate $^{207}\text{Pb}/^{206}\text{Pb}$ ratios for subducted AOC of different ages, as well as AOC bulk rock Cl , K , Nb etc. abundances, then model melting that bulk composition as a garnet pyroxenite and/or eclogite to see if Rairua-like melts can be produced.

Following the suggestion, we modeled the melt from the HIMU source by assuming the eclogite composition (bulk oceanic crust including AOC) after Becker et al. (2000). We assumed large degree (30%) non-modal fractional melting of subducted crust, as its melting degrees should be much higher than the depleted component (MORB mantle; please also see above). We took the Pb isotopic composition of recycled crust after the Pb isotope evolution model by Stracke et al. (2003) assuming subduction at 2 Ga. However, we recognise that there are large uncertainties in model parameters such as chemical composition, source lithology, recycling ages, and hence isotopic compositions. Calculated mixing lines in Figs. 1 and 2 provide an estimate of the required chemical enrichment of the HIMU melt compared to that of the depleted melt. The description on this regard is given in Lines 128-142 in the Supplementary Information.

-The Cl content of melt from the MORB mantle (Table 3) is calculated using the canonical ratio of K/U (12000) and then Cl/K of 0.02 of Shimizu et al (2016). There is much debate about Cl/K ratios. As an alternative value, using a Cl/K ratio of 0.0075 (Saal et al., 2002, *Nature*) would better reproduce the spread seen in Rairua melt inclusions (e.g. Fig. 1). If the authors decide to stay with a Cl/K ratio of 0.02, it would be worth pointing out in the text the range of values present in the literature (0.0075 ± 0.0025 of Saal et al. (2002, *Nature*) to 0.06 ± 0.01 of Kendrick et al. 2017, (*Nature Geoscience*)), and why they chose that particular value. Alternatively the authors could assume Cl behaves perfectly incompatibly during mantle melting, and using the MORB mantle Cl abundances from literature (see my next comment), calculate the melt Cl content in the same fashion as for other trace elements. This method would then not rely on extrapolating across two ratios.

-Table 4, B11. Saal et al. 2002's estimate of MORB mantle is 1.0 ± 0.5 $\mu\text{g}/\text{g}$ using Cl/K , and 0.9 ± 0.63 $\mu\text{g}/\text{g}$ using CO_2/Nb based on a strong CO_2 vs Cl correlation. I recommend that the authors also include the Workman and Hart (2005) Cl estimate of 0.38 ± 0.25 $\mu\text{g}/\text{g}$ for DMM.

As you pointed out, Cl content in the MORB mantle has been a matter of debate. Some previous studies might have overestimated Cl/K in the MORB mantle using the data of MORB affected by melt from small-scale mantle heterogeneity or by contamination. Following your suggestion, we chose Cl concentration in the MORB mantle from the literatures rather than calculating it using canonical K/U value and assuming Cl/K in the MORB mantle. We took Cl in the MORB mantle from Salters and Stracke (2004)'s estimation (0.51 p.p.m.). We believe that this value is reasonable as a model concentration because it is within the range of Cl estimated by other authors using abyssal peridotites combined with volatile/lithophile element ratios (Workman and Hart, 2005), melt inclusions in MORB (Saal et al., 2002), and mantle peridotites (Urann et al., 2017) (please see Supplementary Table 4).

-Table 4, A14. The authors state, “undegassed primitive mantle”, yet their Cl amount (10.4×10^{19} kg) is identical to the “Bulk Silicate Earth” from Sharp and Draper (2013, EPSL). I would use Sharp's BSE terminology, as their study suggests virtually no Cl resides within the core. “Primitive mantle” is an idealized construct.

In response to the comment, we changed the term from ‘Primitive mantle’ to ‘Bulk Earth’. We used the term ‘Bulk Earth’ instead of ‘Bulk Silicate Earth’ because ‘Bulk Earth’ includes Cl not only in the silicate mantle but also in the surface reservoirs (ocean, evaporites, crustal brines) which were supplied from the solid Earth by mantle degassing (Supplementary Table 4).

-Lines 145-149. Subducted Cl is estimated to be $0.6\text{--}1.2 \times 10^{19}$ kg over the past 4Ga, yet Cl is also removed from the mantle during melting, e.g. the calculations in the following paragraph. The critical takeaway (to me) here is that subducted oceanic crust holds far more Cl per mass unit than the MORB mantle- up to 30x more, if one is to believe DMM estimates of peridotite ($0.38\text{--}1 \mu\text{g/g}$ Cl) of Saal et al., (2002), Workman and Hart (2005), etc. This would indicate that subducted AOC contains, as an upper bound, the *majority* of Cl in the mantle ($0.6\text{--}1.2 \times 10^{19}$ kg subducted AOC vs. 0.36×10^{19} kg in DMM using the aforementioned citations). This might be worth noting in the manuscript.

Thank you for the thoughtful comment. In the previous manuscript, we discussed that subducted oceanic crust after dehydration (e.g., HIMU mantle) could be a large

reservoir of Cl in the mantle, but it is also important to explicitly describe that its Cl concentration is much higher than that in the MORB mantle. We revised the text to mention it. Moreover, we added a schematic figure to highlight the significance of subducted oceanic crust as one of major Cl reservoirs in the mantle. Because Cl in the MORB mantle is estimated to be $0.11\text{--}1.4 \times 10^{19}$ kg (Fig. 4 and Supplementary Table 4), Cl delivered by subducted oceanic crust would be equivalent to or even greater than that in the MORB mantle (Lines 182-189).

-Line 152. I believe the paper underestimates the subducted oceanic crust Cl contribution. The 10–20% value appears to be $(0.6 \times 10^{19} \text{ kg} / 5.8 \times 10^{19} \text{ kg} \text{ to } 1.2 \times 10^{19} \text{ kg} / 5.8 \times 10^{19} \text{ kg})$, however the denominator here is the surface reservoir total. Should it not be the estimated mantle Cl total ($4.6 \times 10^{19} \text{ kg}$)? In this case, the range is 13–26%.

We rounded down the number in the previous manuscript, but we showed the number with two significant digits (e.g., 13–26 %) in the revised manuscript (Lines 31 and 194)

-Line 159. The reference here should be the same as the others (superscript).

We showed ‘kg/yr (ref. 9)’ because the number in superscript looks like ‘kg/yr⁹’. We will follow the editorial instruction.

-SIMS measurements. I applaud the authors for the high precision of their SIMS Cl measurements, as indicated by the low 2SE using 10 counting cycles. I would caution that these errors do not shed light on the accuracy of their measurements however. I suggest that machine uncertainty (precision) be propagated with the calibration curve uncertainty (accuracy), perhaps using a bootstrapping method, to provide a meaningful quantification of both accuracy and reproducibility. Extended Data Figure 5 shows the calibration curve used (a single measurement per reference material), however I assume that multiple measurements were made on each reference material, which will contribute to a larger uncertainty than that implied by the stated r^2 value of 0.999.

To clarify the source of uncertainties, we added a detailed description about calibration curves with SIMS in the ‘Method’ section. The measurements were performed on two analytical periods, and we drew a calibration curve before each analytical session. In the previous manuscript, we showed single calibration curve that was taken during one analytical session when the majority of samples (42 out of 50) were measured, for

reference. We did not show a calibration curve taken during the other analytical session because it overlaps with the other. However, in the revised manuscript, we showed two calibration curves in Supplementary Figure 6.

We assessed the machine performance and quality of standard glasses in the previous paper (Shimizu et al., *Geochemical Journal* 51, 299-313, 2017) using the data acquired in 10-month period (from September 2015 to June 2016). We note that the first (November 2015) and second (June 2016) analytical sessions in the present study were performed during that period. In Shimizu et al. (2017), repeated measurements of several kinds of standard glasses showed the reproducibility of $<3\%$ (1 SD). This value is much greater than in-run precision shown in Supplementary Table 1, as you pointed out. In response to the comment, we revised the 'Method' section to clearly describe that the analytical uncertainty for each sample measurement is $<3\%$ given from the repeated measurements of standard glasses (Lines 251-255).

The accuracy of measurements is dependent on the uncertainty of Cl concentrations in glass standards used for calibration lines, as you noticed. The reference Cl values of the international and in-house standards were previously determined by the bulk glass analysis using pyrohydrolysis and ion chromatography (Shimizu et al., 2017) after the method of Shimizu et al. (*Geochemical Journal* 49, 113-124, 2015). Following the uncertainty of bulk Cl analysis determined by these studies, we described that the accuracy of the reported data is estimated to be $<6\%$ (1 sigma). As shown in the Supplementary Data Fig. 5, the calibrations lines acquired in different analytical sessions overlap well. During the 10-month period reported in Shimizu et al. (2017), the slope of the calibration lines ranged from 638 to 666 with 1.5 % relative standard deviation (n=5). Because the change of calibration lines is small between days and we always acquire the calibration line during each analytical session, we believe the variability of calibration lines is not a source of inaccuracy. The description relevant to this point is shown in the Methods (Lines 255-258).

- **Table 1.** I am astonished that duplicate samples have the exact same isotopic, trace element, major element compositions, and associated uncertainties, as that of the duplicated sample. Did the authors independently measure these duplicate samples for all stated values? This would imply perfect reproducibility for EMP, SIMS, and LA-ICP-MS. Please explain this.

It was our mistake to show the ‘duplicates’ in the data table. We made duplicate measurements for hydrogen isotopes, but single measurement for all the other elements/isotopes, on some selected melt inclusions in the early period of this study. For data plotting purpose, we used the same compositions other than hydrogen isotopes in the data file. When reproducing the data table for this manuscript (without hydrogen isotope data), we missed to erase the ‘duplicates’. We corrected the data table in the revised manuscript.

-The Rb content of RAV-08_OL-10_MI-01 is nearly an order of magnitude higher than any other MIs from the suite, while other LILEs do not show such enrichment. While not critical, I wonder if this is a typographical error?

It was not a typo. This melt inclusion has very high incompatible element concentration (Rb, U, Th, Pb) together with high Cl. It was discriminated from the discussion because of very high Cl content (over the threshold of possible Cl assimilation). We suspect such anomalous composition in incompatible elements was also caused by assimilation.

Reply to the comments from Reviewer #2

We are grateful for your careful reading of our manuscript and thoughtful comments and suggestions. Following the suggestions by you and another reviewer, we have revised the manuscript. Below, we reply to your comments and explain how we have revised the manuscript.

Reviewer #2; Overview:

My primary concern, as detailed below, deals with how the authors filter the inclusions based on their Cl/Nb ratios. It appears that they filter the inclusions with high Cl/Nb because they have high Cl/Nb (there is no other apparent geochemical reason for filtering these inclusions). After filtering the high Cl/Nb inclusions, they argue that the inclusions with the highest Cl/Nb that survived the “Cl/Nb filtering process” must represent the Cl/Nb ratio of the HIMU domain. However, the Cl/Nb ratio of the surviving inclusions will depend on the threshold Cl/Nb value that the authors selected for the filtering process. This seems a bit circular. Can't the authors use another geochemical criteria for filtering? Don't the highest Cl/Nb inclusions have some other geochemical signature that can be used for filtering?

There are lots of previous studies on Cl by analyzing submarine glasses and melt inclusions in mantle-derived rocks. However, Cl assimilation from shallow contaminants was always a problem. Because such contaminants are highly rich in Cl compared to other volatiles (i.e., F, S, C) and lithophile elements, only Cl may be affected by assimilation processes in many cases. Seawater or brine assimilation may not always increase water content in the basalts/melts to a detectable level, as uncontaminated basalts originally have some water.

In response to the comment, we assessed the correlation between incompatible elements, including Cl as one of those, in our data set. The melt inclusions from Rairua and Anatonu basalts exhibit a linear trend between incompatible ‘lithophile’ elements (e.g. La versus Nb as shown in Supplementary Fig. 2a in the revised manuscript). Such linear trend is ascribed to partial melting and mixing from two different sources (i.e. HIMU and depleted components), but cannot be caused by seawater or brine assimilation because of low concentration of lithophile elements in such contaminants. If contamination of altered oceanic crust or sediments occurred, correlation between lithophile elements (e.g. La versus Nb) would be blurred, which was not seen in our

data set, except for one melt inclusion which has unusually high concentrations in rare earth elements and large-ion lithophile elements (please see Supplementary Fig. 2a).

When plotting La versus Cl (Supplementary Fig. 2b), the majority of melt inclusions define a linear trend because Cl behaves as an incompatible element like La. However, four melt inclusions in our data set clearly plot above this trend. This fact suggests that Cl concentration is elevated in these four melt inclusions by the processes other than melting and mixing of the primary sources (i.e. HIMU and depleted components). We conclude that high Cl concentration unrelated with lithophile incompatible elements was caused by Cl assimilation.

We gave a threshold at Cl/La = 25 to discriminate melt inclusions that might have been affected by assimilation in terms of Cl from the discussion (4 out of 50). Moreover, one inclusion that has unusually high concentrations in rare earth elements and large-ion lithophile elements (La = 168 p.p.m.), as mentioned above, was not used in the discussion. We described how we filtered out these inclusions in the revised manuscript (Lines 89-105 in the Supplementary Information).

Specific comments:

1. Line 83. The Raivavae host basalts show a bimodal Pb isotopic composition⁸ (Fig. 1a). Older Rairua Formation basalts have more radiogenic Pb, thus having lower $^{207}\text{Pb}/^{206}\text{Pb}$, than younger Anatonu Formation basalts.

Comment: Explain what you means by having “more radiogenic Pb”. Add something like “i.e., higher $^{206}\text{Pb}/^{204}\text{Pb}$ ”. Both ^{207}Pb and ^{206}Pb are “radiogenic”, so this is potentially confusing.

As you pointed out, the sentence in the previous manuscript might cause misleading, because both numerator and denominator are radiogenic (^{206}Pb , ^{207}Pb , ^{208}Pb). We added an explanation to the sentence to tell that ‘radiogenic Pb’ means lower $^{207}\text{Pb}/^{206}\text{Pb}$ and $^{208}\text{Pb}/^{206}\text{Pb}$ in this case (Lines 95-96).

2. Line 86: The Rairua basalts thereby contain more melts from recycled ancient oceanic crust with the HIMU signature than the Anatonu basalts that have an isotopic proximity with the MORB mantle source.

Comment: “The Rairua basalts thereby contain a larger component from recycled oceanic crust...”

We changed the sentence following your suggestion (Lines 96-98).

3. Line 92. This suggests that small-scale chemical heterogeneity is present in the magmas, which is caused by mingling of melts from different mantle sources with varying Pb isotope compositions, as previously documented in melt inclusions in Mangaia basalts (Extended Data Fig. 1)

Comment: The two early studies (Yurimoto et al; Saal et al) found extreme heterogeneity for Pb isotopes in Mangaia melt inclusions. But two later studies (Paul et al; Cabral et al) did not. This should be addressed in the text, as this is a concern for the community.

In response to the comment, we noted that two previous studies reported large Pb isotopic variation in Mangaia melt inclusions, but the other two studies did not find such variation in the revised manuscript (Lines 105-107).

It is no wonder that melt inclusions from Raivavae in this study showed large Pb isotopic variation because two groups of basalts (bulk rocks) with different Pb isotopes have been recognized in this island. Since these melt inclusions plot in between the two groups of basalts in the Pb isotope diagram, we interpreted that such variation was caused by mingling of two types of melts in a magma when melt inclusions were trapped in olivines. On the other hand, Mangaia basalts (bulk rocks) thus far studied do not show large Pb isotopic variation, and therefore the reason for large Pb isotopic variation observed in melt inclusions in Mangaia basalts, if any, is unclear. One possible explanation might be that melts with less-radiogenic Pb isotopes (i.e., melt from depleted source) existed in Mangaia, but has not been discovered in the samples collected on-land. At Tubuai on the same island chain, some basalts with less-radiogenic Pb isotopes were sampled from submarine slopes of the island, while all the subaerial basalts show radiogenic Pb isotopes (Hanyu et al., 2013). We need more studies on Mangaia to test this hypothesis. Another possibility is that there were some technical problems on sample preparation or isotopic measurements in the previous studies for Mangaia, but it is beyond the scope of this paper to argue it..

4. Add melt inclusion sizes to Supp Figure 1. Also, please add photos of all melt inclusions. The authors no doubt have such photos, and I suspect this information will be useful to the community as folks evaluate which types of inclusions, based on

morphology, are most representative.

In response to the comment, we showed sizes of melt inclusions (long and short axes) in Supplementary Table 1 in the revised manuscript. The photos of all melt inclusions were given in a new figure (Supplementary Fig. 5), so readers will see sizes and shapes of melt inclusions visually.

5. Line 96. We determine Cl enrichment by comparing Cl with lithophile elements with similar incompatibility during partial melting (i.e., K and Nb)^{9,14,17} as shown in Figs. 1b and 1c. After filtering out melt inclusions with anomalously high Cl (4 out of 52; Extended Data Fig. 2 and Supplementary Information), we discovered that the Rairua inclusions show higher Cl/K and Cl/Nb ratios than the Anatonu inclusions and normal MORBs, with Cl/K ~ 0.02 and Cl/Nb ~ 517, and that for the Rairua inclusions these ratios are negatively correlated with ²⁰⁷Pb/²⁰⁶Pb.

Comment: This seems circular. See comment below referring to the supplement. After eliminating the high Cl/Nb inclusions, for no clear reason other than the fact that they have high Cl/Nb, the authors argue that one basalt series has higher Cl/Nb than the other. But isn't this a product of the Cl/Nb "threshold value" of 15 that the authors chose? What if the authors chose a lower value, say, Cl/Nb = 5, as their threshold value? This needs to be better justified.

In response to the comment, we gave criteria to discriminate melt inclusions that might be affected by assimilation using Cl/La ratio. Please see our reply to your main comment above for the details.

6. Line 103. Shallow contaminants have less-radiogenic Pb isotopes, thus assimilation of seawater, brine, or brine-impregnated oceanic crust is not responsible for Cl enrichment coupled with radiogenic Pb.

Comment: It is important to show this. Please show all possible contaminants in a plot of ²⁰⁷Pb/²⁰⁶Pb vs ²⁰⁸Pb/²⁰⁶Pb. Show brines, seawater, altered oceanic crust, sediment, etc. Hauff et al. (G-cubed 2003; doi:10.1029/2002GC000421) have some excellent examples of highly altered oceanic crust that would be useful in this regard. All of the possible contaminants should be shown together with the melt inclusions from this study, so that the reader can identify whether the melt inclusions have interacted with these assimilants. This will help the reader understand the arrows in Figure 1, which point to an "assimilation trend". There are so many possible assimilants

that the reader is wondering why the assimilation points up and to the right in the plots?

Thank you for the thoughtful comment and a useful citation. In the revised manuscript, we argued against assimilation by showing Pb isotopic composition of possible assimilants, such as altered oceanic crust and sediments given by Hauff et al. (2003) and seawater and seawater-derived brines by Frank (2002). We displayed Pb isotopic ranges of them in the isotope diagram (Fig. 1a). Bulk oceanic crust, sediments, seawater and brines have Pb isotopic composition similar to or less-radiogenic (higher $^{207}\text{Pb}/^{206}\text{Pb}$) than the Anatonu melt inclusions. Therefore, their assimilation do not account for the Cl increase coupled with radiogenic Pb (low $^{207}\text{Pb}/^{206}\text{Pb}$) observed in Rairua melt inclusions. This discussion is added to the main text in the revised manuscript (Lines 115-118).

We should also mention that altered veins that bear secondary minerals (possibly rich in Cl) may have very radiogenic Pb isotopes because they have elevated U/Pb and Th/Pb, as reported by Hauff et al. (2003). If such vein materials selectively assimilated to magmas, the melts might have high Cl coupled with radiogenic Pb isotopes. However, the Pb isotopic trend defined by the Rairua, Anatonu, and Mangaia melt inclusions do not point to the vein materials in $^{207}\text{Pb}/^{206}\text{Pb}$ – $^{208}\text{Pb}/^{206}\text{Pb}$ space (Supplementary Fig. 1b), because radiogenic Pb isotopes in vein materials are the product of recent radiogenic ingrowth. Consequently, assimilation of vein materials cannot be responsible for radiogenic Pb isotopes associated with elevated Cl observed in Rairua and Mangaia melt inclusions. We added detailed discussion on it and a new figure (Supplementary Fig. 1b) to the revised manuscript (Lines 107-120 in Supplementary Information).

7. Line 104. Despite small Pb isotopic variation, the Mangaia melt inclusions also show high Cl/K and Cl/Nb 19,

Comment: Actually, Yurimoto et al, and Saal et al., found enormous Pb isotopic variability at Mangaia. This is a concern.

The sentence was rephrased to avoid confusion (Lines 118-119). Regarding the Pb isotopic variability reported by Saal et al. and Yurimoto et al., please see our reply to your main comment above.

8. Line 104. Despite small Pb isotopic variation, the Mangaia melt inclusions also show high Cl/K and Cl/Nb 19, which overlaps that of the Rairua inclusions. Chlorine

enrichment should thus be a feature of the HIMU mantle source.

Comment: In Fig 1, a couple of the Mangaia inclusions show low Cl/K and low Cl/Nb, but have the same Pb isotopic compositions as the other Mangaia inclusions. It would be good to mention this. Could it be possible that the low Cl/Nb inclusions from Mangaia are the most “primitive” (i.e., least contaminated)?

The melt inclusion with the lowest Cl/K and Cl/Nb in Mangaia shows low Cl (249 p.p.m.), high P (5.56 wt%), and low U (0.296 p.p.m.) relative to the other inclusions (Cabral et al., 2014). There might be some reasons for an unusual chemical composition on this melt inclusion, but it is beyond the scope of this manuscript to discuss compositional variation among Mangaia melt inclusions in the previous study.

9. Line 108 [Note: in Supplementary Information]. The isotopic variation defined by Raivavae host basalts is best explained by binary mixing of melts from HIMU and MORB mantle sources

Comment: There is another depleted components in the Cook-Australis. Konter et al. (epsl, vol 275, 2008) showed a pervasive “C” component throughout the Cook-Australis. Why do the authors prefer a MORB component over the already-identified C component? This is worth mentioning.

Thank you for reminding us the literature. We noted that the depleted component may be either ambient mantle (Lassiter et al., 2003) or C component (Konter et al., 2008) in the discussion in the supplements (Lines 143-148 in Supplementary Information).

10. Line 112 [Note: in Supplementary Information]. We calculated the composition of the MORB mantle melt assuming 1% fractional melting of the MORB mantle with bulk partition coefficients between melt and garnet peridotite

Comment: Why do the authors use such a low melt fraction? Please explain (in the text) why such a low value is selected.

We assessed if the melting/mixing model reproduces the elemental ratios (Cl/K, Cl/Nb, K/U, and Nd/Hf) of the melt inclusions by changing the melting degree of the depleted source (MORB mantle). Because the elements studied here are incompatible, the mixing trajectories are not largely modified if we assume small (1%) and large (5%) degree of partial melting. The reason why we assumed low-degree melting here is based on our previous study (Hanyu et al., 2013). We found basalts with less-radiogenic Pb isotopic

composition (like Anatonu basalts) from submarine slope of Tubuai Island, the neighboring island of Raivavae, while Tubuai subaerial basalts have radiogenic Pb isotopes (like Rairua basalts at Raivavae). We reproduced composition of various trace elements (i.e., spidergram) of the basalts with less-radiogenic Pb by mixing the basalts with radiogenic Pb (i.e., HIMU melt) and melt from the depleted component. In this model, we showed that low-degree partial melting of the depleted component is required to reproduce the heavy rare earth element composition. We believe that such low-degree melting was achieved if the depleted component (ambient mantle) was heated near the outer edge of the mantle plume. Moreover, it is suggested (e.g., Stracke, *Chemical Geology*, 2012) that the depleted peridotitic component would melt at small degree because of the high solidus temperature relative to the recycled eclogite/pyroxenite/fertile peridotite. We recognise that there are uncertainties in chemical compositions, mineral modes and melting degrees of the source components, but for the purpose of model calculation, we assumed 1% non-modal fractional melting of the depleted component (MORB mantle). We described this point in the supplements (Lines 143-154 in the Supplementary Information).

11. Cl/Nb ratios in the most HIMU glasses (that is, the 3 most HIMU lavas from Dredge 10) examined by Jackson et al (G-cubed 2015) are 12.8–13.7. These values are inconsistent with the hypothesis put forward by the authors here.

The HIMU glasses from Tuvalu Island studied by Jackson et al. (2015) have Cl/Nb (12.8– 13.7) which is well within the range of Cl/Nb of Mangaia and Rairua melt inclusions. These Tuvalu glasses have $^{207}\text{Pb}/^{206}\text{Pb}$ between that of Mangaia and Rairua. Therefore, the Tuvalu glasses plot in between the ranges defined by Mangaia and Rairua melt inclusions in $^{207}\text{Pb}/^{206}\text{Pb}$ –Cl/Nb space. We suggest in the present paper that the similar Cl/Nb but different Pb isotopes of Mangaia and Rairua might be ascribed to different formation ages of the HIMU source(s). Following this theory, the HIMU source of the Tuvalu basalts might have had intermediate formation age between that of the Mangaia and Rairua (Raivavae) basalts.

12. Figure 3. It would be useful to add fields for carbonated peridotite and carbonated eclogite/pyroxenite experimental compositions. As it stands, the authors show data trending outside of the “peridotite melting” field, but this doesn’t really illuminate the authors’ preferred model of carbonate-rich melting.

Thank you for the constructive comment. In the revised manuscript, we changed the plots from the SiO₂-CaO space to the SiO₂-CaO/Al₂O₃ space in the revised Fig. 3, because the latter is more widely used in the previous literatures. In this figure, we showed the compositional ranges of experimental silicate and carbonated melts from not only peridotite but also eclogite (silica-rich and silica-deficient) by referring to Hirschmann et al. (2003), Gerbode and Dasgupta (2010), Dasgupta et al. (2010) and Jackson and Dasgupta (2008). In addition to the previous argument that carbonate-free peridotite cannot generate low-SiO₂ (<45 wt%) alkalic melts, we discussed other possible sources. Carbonate-free silica-deficient eclogite/pyroxenite has been extensively studied and is suggested to be a potential source of low-SiO₂ melts, although the experimental melts have lower CaO/Al₂O₃ (because of too aluminous composition) at low SiO₂. We described accordingly that carbonated sources are more likely to account for the major element compositions of the low-SiO₂ melt after the discussion by Dasgupta et al. (2010) and Jackson and Dasgupta (2008) (Lines 124-130).

13. Supplement. We did not find any highly Cl-enriched melt inclusions equivalent to type-3 inclusions in our sample sets. However, a small number of inclusions in the Rairua and Anatonu basalts show Cl enrichment that deviate from the compositional trend defined by the majority of inclusions, which was presumably caused by brine assimilation. A single Rairua inclusion had much higher Cl/Nb ratio (~53) than other inclusions (Extended Data Fig. 2a). Three Anatonu inclusions had high Cl/Nb (>15) that define a steep trend in Cl versus Cl/Nb space. This excessively high Cl/Nb is not due to Nb depletion because these inclusions show higher Cl/Nb than the others for a given 1/Nb (Extended Data Fig. 2b). We excluded these four melt inclusions and also some inclusions in the literature (shown by open symbols in Extended Data Fig. 2) from the discussion as they were likely contaminated by secondary brines.

Comment: The authors exclude the high Cl/Nb inclusions because they have high Cl/Nb? Is there another geochemical indicator in these high Cl/Nb inclusions that can be used?

In response to the comment, we gave criteria to discriminate melt inclusions that might be affected by assimilation using Cl/La ratio. Please see our reply to your main comment above for the details.

14. Line 108. The isotopic variation defined by Raivavae host basalts is best explained by binary mixing of melts from yHIMU and MORB mantle sources^{8,43,44}.

Comment: This seems circular, somehow. By excluding the lavas with the highest Cl/Nb (i.e., >15), the authors then conclude in the manuscript that HIMU lavas have Cl/Nb of about 15. Is there some other geochemical indicator that the authors can use to exclude the highest Cl/Nb inclusions? Do the high Cl/Nb inclusions have some other bizarre or strange trace element (Rb/Nb? K/Nb?) or major element (K/Ti?) compositions? It just seems strange to filter using an arbitrary cutoff for Cl/Nb to then conclude that there is a specific Cl/Nb that is “primitive”, as this “primitive ratio” will always be determined based on the arbitrary cutoff used by the authors.

In response to the comment, we gave criteria to discriminate melt inclusions that might be affected by assimilation using Cl/La ratio, as replied above. We did not find any bizarre lithophile element compositions and ratios that are tied to ‘anomalous’ Cl enrichment. This is because contaminants such as seawater and brine are highly rich in Cl but not always in lithophile elements. Only example that might show a possible modification of lithophile element composition coupled with brine (or brine-impregnated oceanic crust)-derived Cl is the sample RAV-8 OL-10 MI-01, but the other melt inclusions with anomalous Cl have normal lithophile element compositions.

15. Supplementary Figure 3: Try plotting Ca/Al instead of Ca in the lower panel. This usually tends to show better correlations with Pb isotopes, and is also interpreted to reflect a role for carbonatite in the melt source (see Jackson and Dasgupts).

Thank you for the thoughtful suggestion. We changed the plots from the SiO₂-CaO space to the SiO₂-CaO/Al₂O₃ space in the revised Fig. 3. The plots for ²⁰⁷Pb/²⁰⁶Pb versus CaO/Al₂O₃ are given in Supplementary Fig. 3. We reinforced the argument on the major elements, as replied above.

16. Supplementary Table 2. Paul et al and Cabral et al found very little Pb isotopic variability in Mangaia melt inclusions, which contrasts with extreme Pb isotopic variability in the melt inclusions from Mangaia identified by Saal et al and Yurimoto et al. This is a big concern in the community, particularly because Cabral et al worked on olivine-hosted inclusions from two of the same basalts examined by Saal. Therefore, it is critical to document standard and unknown analyses to ensure that any Pb isotopic variability is real, and not an analytical artifact. The following should be addressed explicitly in the supplement to help readers assess the data:

A. It appears that the Pb isotopic standard was run only 5 times during the analytical

session?

B. Over how many days were the melt inclusions analyzed? How many analyses of the Pb isotopic standard were made during each day that melt inclusions were analyzed?

We showed five repeated measurements of BHVO-2G in the previous manuscript. We also conducted five repeated measurements of BCR-2G during the analytical session. In the revised manuscript, we presented measured values of both standards in the Supplementary Table 2. As noted in the Supplementary Information, we performed Pb isotope analysis on two separate days. BHVO-2G and BCR-2G were measured each day prior to analysing unknown samples, as shown in a new figure (Supplementary Fig. 7). Eight and 42 successful isotope measurements were made on 10 December 2015 and 26 December 26, respectively.

C. Documenting the analytical session. A plot of $^{207}\text{Pb}/^{206}\text{Pb}$ and $^{208}\text{Pb}/^{206}\text{Pb}$ vs time, showing melt inclusions and standards, would be extremely useful. Include in run error bars for each analysis, and document different analytical sessions in the plot.

In response to the comment, we added a new figure in which the isotopic ratios of the bracketing standards, international standards, and unknown samples (before correction using bracketing standards) are displayed in the analytical order (i.e., time) (Supplementary Fig. 6). The error bars correspond to 2 standard errors. The measurements were performed on two separate days. In the beginning of the analytical session of each day, the international standards (BCR-2G and BHVO-2G) were measured for five times. The bracketing standards were measured before and after each measurement of international standards and unknown samples (i.e., melt inclusions) for the purpose of correcting external mass bias, but they also demonstrate the machine stability during the analytical sessions. The Pb isotopic variation observed among the studied melt inclusions is significantly greater than the drift of Pb isotope ratios of bracketing standards during the analytical sessions (Lines 296-303 in the Method section).

D. Did the laser spot overlap with phase boundaries? Olivine-melt inclusions, or spinel-melt inclusions. Cabral et al found that, whenever the beam overlapped with phase boundaries, they would find Pb-isotopic variability that overlapped with the extreme inclusions identified by Saal et al. It is critical to avoid analyses where the beam overlaps with phase boundaries.

We avoided cracked parts, former shrinkage bubbles, spinels and edges of melt inclusions for 48 samples out of 50 reported in the present manuscript. For two melt inclusions (samples RAV-08 OL-10 MI-01 and RAV-20 OL-07 MI-01), we did not have enough space remaining for Pb isotope measurements, and thereby the laser spot overlapped the surrounding host olivines. However, by monitoring the Pb isotope ratios during the sample run, we did not find any unusual drift of isotope ratios. As you mentioned, the condition of spattering and ionization may be affected when primary beam overlaps phase boundaries during the SIMS measurements. We believe that such effect may be much smaller during laser ablation with ICP-MS. However, in response to the analytical concern, we explicitly described in the ‘Method’ section that the laser spot for Pb isotope analysis overlapped the surrounding olivines for the two melt inclusions (Lines 284-288 in the Method section).

E. Paul et al found that surface contamination on the samples was rich in Pb, and they had to carefully identify and eliminate the data from the first, Pb-rich, pulse of data at the start of each laser analysis. This surface contamination issue is critical to avoid, and should be discussed in the supplement.

We also cared possible surface contamination. In our analytical program, we performed pre-ablation for 10 seconds to remove surface contaminants. After starting ablation, data from the first several cycles are discarded until the beam intensity is stabilized. Any additional contaminants may be also taken away at this stage. In case when unexpected contaminants appeared during measurements, we monitored the signal intensity and isotope ratios during each sample run. We described this issue in the ‘Method’ in the revised manuscript (Lines 288-291). The detailed description on the analysis are given in our previous publication (Chang et al., 2014; Kimura et al., 2016), which are referred to in the ‘Method’ section.